# Healthy Lifestyle Intervention and Weight Loss Improve Cardiovascular Dysfunction in Children with Obesity

**DOI:** 10.3390/nu13041301

**Published:** 2021-04-15

**Authors:** Giulia Genoni, Veronica Menegon, Alice Monzani, Francesca Archero, Francesco Tagliaferri, Valentina Mancioppi, Caterina Peri, Simonetta Bellone, Flavia Prodam

**Affiliations:** 1Neonatal and Pediatric Intensive Care Unit, Maggiore della Carità University Hospital, 28100 Novara, Italy; genonigiulia@gmail.com; 2Vascular Surgery, Maggiore della Carità University Hospital, 28100 Novara, Italy; menegonveronica@yahoo.it; 3Division of Pediatrics, Department of Health Sciences, University of Piemonte Orientale, 28100 Novara, Italy; alice.monzani@gmail.com (A.M.); francesca.archero@gmail.com (F.A.); cecco.taglia@gmail.com (F.T.); valentinamancioppi@gmail.com (V.M.); cate.peri@gmail.com (C.P.); simonetta.bellone@med.uniupo.it (S.B.); 4Interdisciplinary Research Center of Autoimmune and Allergic Diseases, University of Piemonte Orientale, 28100 Novara, Italy; 5Department of Health Sciences, University of Piemonte Orientale, 28100 Novara, Italy

**Keywords:** pediatric obesity, cardiovascular dysfunction, behavioral intervention, metabolic syndrome, cardiovascular prevention keyword

## Abstract

Background: The aim of this study was to determine the effects of a 12-month healthy lifestyle intervention based on diet plus physical activity on cardiovascular structure and function in children and adolescents with obesity; Methods: In this longitudinal study we assessed changes in anthropometric, biochemical and cardiovascular variables in 55 subjects with obesity (6–16 years) before and after a 12-month behavioral program based on Mediterranean diet plus exercise regimen. Subjects were divided in two groups based on negative change in BMI z-score ≥10% from baseline: weight losers (WL) and non-weight losers (NWL); Results: After 12 months, WL showed a significant improvement of metabolic parameters. Treatment was effective in increasing the mitral peak early diastolic velocity E and the E/A ratio. In subjects with a reduction of the number of NCEP-ATPIII metabolic syndrome criteria, lifestyle intervention reduced left ventricular area and volume. Intervention reduced carotid intima-media thickness in subjects showing a decrease of the systemic blood pressure; Conclusions: In children with obesity, cardiovascular impairment could be partially reversed by a healthy lifestyle intervention. To adopt prompt behavioral programs in childhood obesity is crucial both for prevention and treatment of precocious complications and could have an exponential impact on long-term morbidity and mortality.

## 1. Introduction

The rising prevalence of childhood obesity worldwide is an important public health issue. Pediatric obesity is associated with a broad range of severe complications, increasing the risk of premature morbidity and mortality [1]. Due to the pediatric obesity epidemic, increased incidence of coronary heart disease and cardiovascular deaths is expected in young adulthood. By 2035, it is estimated that the prevalence of coronary heart disease will increase from 5 to 16%, with more than 100,000 excess cases attributable to obesity [2]. Adolescents with obesity have a hazard ratio of 4.9 for death in early adulthood for coronary heart disease, of 2.6 for stroke, 2.1 for sudden death, and 3.5 for death for total cardiovascular causes [3]. Furthermore, even during childhood, obesity impairs cardiac and vascular structure and function [4,5,6] and some metabolic factors such as insulin resistance, serum uric acid and the presence of metabolic syndrome (MetS) might play an adverse role in heart remodeling in children and adolescents [4]. Given this alarming data, a prompt and aggressive therapeutic strategy in children with obesity is mandatory in order to reverse modifiable obesity-related risk factors, to prevent cardiovascular dysfunction, and ultimately to reduce cardiovascular deaths. The treatment of pediatric obesity is mainly based on exercise, dietary, and behavioral interventions. However, little is known about the reversibility of cardiac abnormalities in children and adolescents with obesity undergoing a lifestyle intervention program.

Therefore, the purpose of this study was to determine the effects of weight loss, after a 12-month healthy-lifestyle intervention based on diet plus physical activity, on cardiovascular structure and function in children and adolescents with obesity.

## 2. Materials and Methods

### 2.1. Study Design and Population

We performed a single-center longitudinal study at the Pediatric Endocrinology Unit, Department of Health Sciences, University of Piemonte Orientale, Novara, Italy. We enrolled from December 2016 to February 2018 subjects aged 6–16 years if they had obesity according to the International Obesity Task Force (IOTF) criteria [7], and not on a weight-loss diet. Exclusion criteria were endocrine or genetic obesity, type 1 or type 2 diabetes, previous heart, respiratory, liver and kidney diseases, current or past use of hormonal or interfering therapies (lipid-lowering, hypoglycemic, or antihypertensive treatments). The protocol was conducted in accordance with the declaration of Helsinki and the Local Ethic Committee approved the study (CE 95/12) that was registered on clinicaltrial.gov (NCT03169257), accessed on 15 April 2021. Informed written consent was obtained from all subjects’ parents.

### 2.2. Intervention

Subjects were evaluated at baseline (T0) and after 12 months (T12) of a lifestyle intervention program. A trained pediatric endocrinologist and a nutritionist assessed the habitual diet and administered an isocaloric Mediterranean balanced diet (50–55% carbohydrates with less than 12% of sugars; 16–17% proteins; 28–32% fats with less than 10% saturated fats; fibers 8.5 g/1000 Kcal) according to Italian LARN Guidelines for age and gender [8]. Mediterranean food pyramid was explained together with the diet [9]. Two or more servings of fish and legumes per week and five of raw or cooked vegetables and fruits per day were recommended. Daily intake of nuts and extra-virgin olive oil was suggested, whereas that of processed foods, bakery, candies, trans-fats, and sugars was discouraged. Pictures of dishes were presented on booklets to each subject and their parents to educate to portion size and Kilocalories. Counseling was also focused on helping increase the palatability of avoided foods. Moreover, subjects underwent a physical activity regimen. It was suggested to perform daily session of aerobic training for 45–60 min per day for 7 days per week. The type of training was explained by the pediatric endocrinologist and included aerobic activities fast walking, running, ball games, or swimming. Subjects could choose the type of activity based on their preference and their age. At T12, subjects were divided in two groups on the basis of the negative change in BMI z-score ≥10% from baseline: a weight losers group (WL) and non-weight losers group (NWL) [10,11].

### 2.3. Anthropometric and Biochemical Variables

At baseline and at T12, we evaluated anthropometric and biochemical variables. Height, weight, waist circumference (WC), systolic (SBP) and diastolic (DBP) blood pressure were measured as previously described [4] and body mass index (BMI) was calculated. Pubertal stages were evaluated according to Tanner criteria. 

In all subjects, after a 12-h overnight fast, blood samples were taken for measurement of: glucose (mg/dL), insulin (μUI/mL), total cholesterol (mg/dL), high-density lipoprotein-cholesterol (HDL-c, mg/dL), triglycerides (mg/dL), serum uric acid (sUA) (mg/dL), using standardized methods in the Hospital’s Laboratory4. Low-density lipoprotein-cholesterol (LDL-c) was calculated by the Friedwald formula. sUA (mg/dL) was measured by Fossati method reaction using uricase with a Trinder-like endpoint. Study subjects also underwent at baseline and T12 an oral glucose tolerance test (OGTT) (1.75 g of glucose solution per kg, maximum 75 g) and samples were drawn for the determination of glucose and insulin every 30 min for 2 h. Insulin resistance was calculated using the formula of homeostasis model assessment (HOMA)-IR. Insulin sensitivity at fasting and during OGTT was calculated as the formula of the Quantitative Insulin Sensitivity Check Index (QUICKI) and Matsuda index (ISI). Glucose was expressed in mg/dL (1 mg/dL = 0.05551 mmol/L) and insulin in μUI/mL (1 μUI/mL = 7.175 pmol/L) in each formula. Impaired fasting glucose (IFG) and impaired glucose tolerance (IGT) were defined according to American Diabetes Association [12] and MetS by using the modified National Cholesterol Education Program/Adult Treatment Panel III (NCEP-ATP III) criteria of Cruz and Goran [13].

### 2.4. Echocardiographic and Vascular Assessment

Transthoracic echocardiogram was performed at baseline and after the 12-month lifestyle intervention, using a Vivid 7 Pro ultrasound scanner (General Electric Healthcare, USA) by a sonographer and the images were reviewed by an expert pediatric cardiologist, blinded to subjects’ clinical data. Measurements of left ventricle (LV end-diastolic diameter, LVEDD; LV end-systolic diameter, LVESD; interventricular septum at end diastole, IVSD; LV posterior wall at end diastole, LVPWD), relative wall thickness (RWT), left atrium diameter (LAD), the maximum LA volume, and LV ejection fraction were obtained according to established standards [14]. LV mass (LVM) was derived from the Devereux formula and indexed to body surface area (left ventricular mass index (LVMI). Using pulsed wave Doppler, mitral inflow velocities, peak early diastolic velocity (E), peak late diastolic velocity (A), E/A ratio, were measured. Pulsed wave tissue Doppler of the lateral mitral annulus was used for the measurement of early peak diastolic mitral annular velocity (E’). The E/E’ ratio was calculated.

Vascular measurements were performed with a high-resolution ultrasonography (Esaote MyLab25TM Gold, Esaote, Italy) using an 8 mHz linear transducer and a 5 mHz convex transducer for the abdominal aorta, by an expert sonographer and images were then reviewed offline by an expert vascular surgeon blinded to subjects’ clinical status. Carotid artery intima-media thickness (CIMT), abdominal aortic diameter at maximum systolic expansion (Ds) and minimum diastolic expansion (Dd), brachial artery diameters, brachial artery peak systolic velocity (PSV) and end diastolic velocity (EDV) were measured as previously described and aortic strain (S), pressure strain elastic modulus (Ep), pressure strain normalized by diastolic pressure (Ep*) and brachial artery flow-mediated dilation (FMD) were calculated. S is the mean strain of the aortic wall, Ep and Ep* are the mean stiffness [15]. The brachial artery maximum diameter recorded following reactive hyperemia was reported as a percentage change of resting diameter (FMD = peak diameter−baseline diameter/baseline diameter) [16].

### 2.5. Statistical Analysis

All data are expressed as mean ± standard deviation (SD), absolute values and percentages, as appropriate. A sample of 15 individuals has been estimated to be enough to demonstrate a difference of 10% in LV diameter with a SD of 0.44 cm with 90% power and a significance level of 95% in the Student t-test between weight losers and non-weight losers according to published data [4]. Skewed variables were log transformed. The Wilcoxon signed-rank test was used to assess changes in cardiovascular variables. A two-way repeated measure ANOVA was performed to evaluate the time effect, the treatment effect and the interaction effects of: the negative change in BMI z-score ≥10%, any reduction of HOMA-IR, of the number of MetS criteria according to NCEP-ATPIII classification [13], of SBP and of DBP on the dependent variables (cardiovascular parameters). Sum of squares type III was used. The following covariates were also subsequently introduced: sex and pubertal status. Statistical significance was determined at a *p*-value of <0.05. All the statistical analyses were performed using R Statistical Software and SPSS for Windows version 25.0 (IBM Corp., Armonk, NY, USA). 

## 3. Results

Of the 80 subjects who underwent baseline evaluation [4], 62 agreed to participate to the longitudinal study while 18 refused to sign the consent and were excluded. During the study protocol, 4 participants (6.5%) were lost at follow-up and some data were missing in 3 subjects (4.8%), and thus they were excluded from the final study population. Out of 55 subjects, 28 (51%) were considered WL and 27 (49%) NWL based on a negative change in BMI z-score ≥10% from baseline (Figure 1).

The baseline characteristics were similar in the WL and NWL group and are shown in Table 1.

At baseline, 11 (39.3%) and 13 (48.1%) subjects matched the NCEP ATPIII criteria for MetS, in the WL and NWL group, respectively. In WL subjects, 2 (7.2%) children were positive for one component of MetS, 14 (50.0%) for two components, 9 (32.1%) for three components, and 3 (10.7%) for four components. In the NWL group, 5 (18.5%) subjects were positive for one component of MetS, 9 (33.4%) for two components, 6 (22.2%) for three components, 6 (22.2%) for four components, and 1 (3.7%) for all the five components of MetS.

### Echocardiographic and Vascular Variables

After the 12-month lifestyle intervention, in WL subjects, we found a decrease of the heart rate (*p* < 0.01) and CIMT (*p* < 0.01) and an increase of left atrial area (*p* < 0.04), mitral peak early diastolic velocity (E, *p* < 0.001), and abdominal aortic diameter at minimum diastolic expansion (*p* < 0.03) at T12 compared to baseline values.

NWL subjects had higher left ventricular end-diastolic diameter (LVEDD, *p* < 0.04) at T12 compared to T0. Even in this group, an increase of the E (*p* < 0.007) and a decrease of the CIMT (*p* < 0.01) was shown after 12 months (Table 2).

A two-way repeated measure ANOVA was performed to evaluate the time effect, the treatment effect and the interaction effect on all the cardiovascular dependent variables according to the negative change in BMI z-score ≥10% from baseline. We found a significant effect of the interaction (time×treatment) for the mitral peak early diastolic velocity E (F:4.562, *p* < 0.04) and E/A ratio (F:5.614, *p* < 0.02), even when adjusted for sex and pubertal stage. We did not find any significant effect of the interaction analyzing subjects who did and did not show a reduction of HOMA-IR. Out of 55 subjects, 20 (36.4%) showed a reduction in the number of metabolic syndrome criteria based on NCEP-ATPIII classification. Reducing the number of MetS criteria, LV area and volume decreased with a significant effect of the interaction (LV area, F:5.918, *p* < 0.01; LV volume, F:3.863, *p* < 0.05), even when adjusted for sex and pubertal stage (Figure 2A).

By analyzing single components of MetS, 24 (43.6%), subjects showed a decrease of the SBP and 20 (36.4%) of the DBP. In children who presented a reduction of SBP or DBP, a significant effect of the interaction was shown for CIMT (SBP, F:3.940, *p* < 0.05; DBP, F:7.988, *p* < 0.007), even weighted for confounding factors (Figure 2B).

## 4. Discussion

In this preliminary study, we found an improvement of cardiovascular dysfunction in children with obesity 12 months after an intervention based on healthy lifestyle. 

Lifestyle interventions targeting healthy eating, improving exercise, and reducing sedentary activity are the cornerstones therapies of obesity in childhood and adolescence. In adults, lifestyle and behavioral interventions showed a weight-loss efficacy of 5–10%, often resulting in CV risk factors improvements [17,18]. Furthermore, pediatric lifestyle intervention trials have also reported improvements in body composition and metabolic parameters [19,20,21]. In line with this, we also found that children with a reduction in BMI z-score from baseline, improved glucose metabolism, HDL-cholesterol and insulin sensitivity with a parallel reduction of insulin resistance.

In a previous study, we found that children and adolescents with obesity had slightly impaired cardiovascular structure and function compared with normal-weight subjects [4], even if cardiovascular parameters fall all in the normal ranges. This highlights that cardiac remodeling and impaired vascular function in pediatric subjects with obesity is difficult to ascertain with standard imaging techniques and without a control population. 

As the direct cardiovascular effects of childhood obesity are relatively mild and subtle, it is also difficult to measure the effect of intervention programs on cardiac remodeling and cardiovascular function. In the present study, we found a significant increase of the E/A ratio (mainly related to the increased early diastolic velocity/E-velocities). The effect of weight loss on diastolic function in children has been previously reported [22,23,24]. In a study by Ippisch et al., adolescents with morbid obesity aged 13–19 years undergoing bariatric surgery after 10 months showed a decrease of mitral A-velocities, and an increase of the E/A ratio and of mitral E’-velocities parallel to BMI reduction [23]. In addition, an exercise-training program of 8 weeks improved diastolic function (mitral inflow A-velocities decreased, mitral Doppler tissue derived E’- and A’-velocities increased, and E/E’ ratio decreased) without significant changes of BMI in a group of children with obesity aged 8–14 years [23]. Ingul et al. recently performed a randomized controlled trial in 99 adolescents with obesity and reported that a 12-week aerobic interval-training program coupled with nutrition advice was effective in improving systolic and diastolic cardiac function [24]. Furthermore, a low carbohydrate diet for 6 months reduced BMI and improved right ventricle diastolic function [25]. Taken together, these data suggest that the diastolic dysfunction may be reversed by weight loss. Moreover, previous studies have shown that a reverse heart remodeling occurs after the normalization of functional changes [26]. These findings outline the importance of a long-term follow-up of patients with obesity and of strategies aiming at preventing the drop-out of lifestyle intervention programs.

An interesting finding of the current study is the significant effect of metabolic syndrome on left ventricular area and volume. In children with a reduction of the number of MetS criteria after the 12 months intervention, LV area and volume significantly decrease compared with subjects with the same or a greater number of MetS criteria from the baseline. As previously shown, in children with obesity, MetS was associated with greater heart dimension and mass with straightforward linear raises when increasing the number of matched criteria for MetS [4]. Subjects who met the MetS criteria presented the worst metabolic parameters (dyslipidemia, dysglycemia), with higher prevalence of hypertension and altered BMI, with a significant impact on cardiac structural alterations. Reducing the number of MetS criteria, as reducing CV risk factors, could change heart remodeling. Our data confirm the influence of MetS also in the pediatric age, suggesting the usefulness of programs aimed at preventing or reversing MetS and the related future cardiovascular dysfunction.

By evaluating single MetS criteria, we found that CIMT significantly decreased along with the reduction in BP. Systemic hypertension is a common complication of childhood obesity. Elevated BP has been related to endothelial and smooth muscle cell dysfunction, arterial stiffness, and increased left ventricular mass [27]. Systemic hypertension in children with obesity might be due to impaired endothelial function or activation of the sympathetic nervous system or insulin resistance [27]. In adults, a reduction of 5 mmHg in diastolic BP is associated with a 35% decrease of stroke [28]. Furthermore, in adults, 1 SD increase in CIMT has been associated with a 2-fold raised risk of ischemic stroke or myocardial infarction; thus, changes of CIMT could determine important health gains at a population level. Farpour-Lambert et al. showed, in a randomized controlled trial enrolling 44 prepubertal children with obesity that a 3-month exercise-training program significantly reduced systolic and diastolic BP and improved arterial stiffness and CIMT [29]. Other pediatric studies have demonstrated improvements in flow-mediated dilatation and intima-media thickness with diet alone, exercise alone or diet and exercise [10]. Notably, we also found a reduction in CIMT in non-weight losers subjects, suggesting that being in a behavioral intervention project per se could have beneficial effects on vascular parameters, even in the absence of a significant weight loss. 

Our study has several potential limitations. The major limit is the relatively small sample size. However, a long-term follow-up during lifestyle intervention programs is a challenge also in pediatrics, as recently shown [19]. Furthermore, regarding the intervention, we did not monitor the intensity and the effective duration of physical activity. The use of telemetric methods like smart watches could be helpful for monitoring the subjects’ compliance. Moreover, it was not a randomized controlled trial. We performed a perspective data collection, which may be susceptible to selection bias. Finally, a more extensive use of vascular imaging modalities including speckle tracking echocardiography and cardiac MRI would have certainly improved the results of the current study.

## 5. Conclusions

In conclusion, this preliminary study shows that a healthy lifestyle intervention could partially reverse cardiovascular dysfunction in children with obesity. This effect is probably mediated by several mechanisms like the reduction of body weight, the improvement of metabolic status and the reduction of systemic blood pressure. In fact, children undergoing weight loss showed an improvement of diastolic heart function, a reduction of the number of MetS criteria was accompanied by a decrease of left heart volumes, and a reduction of systemic blood pressure determined an amelioration of CIMT. 

On one hand, to prevent or reverse overweight and obesity in childhood should be a main goal of pediatricians and pediatric societies, thus blocking the cascade of precocious heart-adverse remodeling cardiovascular disease future. In this view, the pediatric age is a window of opportunity for improving public health. On the over hand, the evaluation of cardiovascular risk and risk stratification should become common practice during a pediatric visit for children with obesity [30]. 

Finally, further larger studies and randomized trials are warranted to confirm our preliminary findings, and future research should investigate the molecular basis of these changes.

## Figures and Tables

**Figure 1 nutrients-13-01301-f001:**
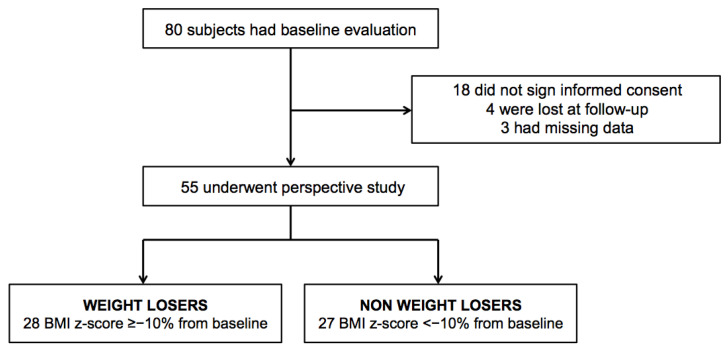
Longitudinal study flow diagram.

**Figure 2 nutrients-13-01301-f002:**
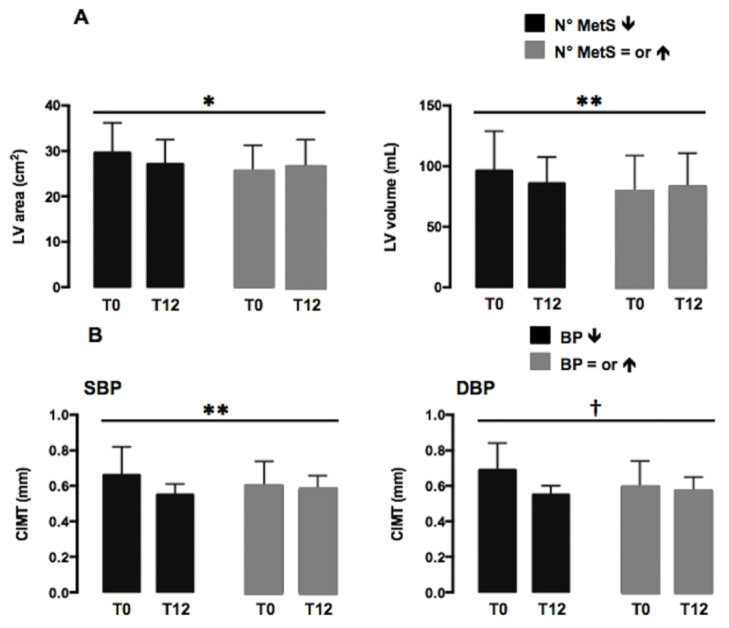
(**A**) Variations of LV area and volume after a 12-month healthy lifestyle intervention in subjects with (black bars) or without (grey bars) a reduction of the number of metabolic syndrome (MetS) criteria according to NCEP-ATPIII classification [17]. (**B**) Variations of CIMT after a 12-month healthy lifestyle intervention in subjects with (black bars) or without (grey bars) a reduction of systolic (SBP) and diastolic (DBP) blood pressure. Data are expressed as marginal mean ± SEM. Data are significant in interaction (* *p* < 0.01; ** *p* < 0.05; † *p* < 0.007). T0: baseline; T12: after 12 months of behavioral intervention.

**Table 1 nutrients-13-01301-t001:** Baseline characteristics of the weight losers (WL) and non-weight losers (NWL) subjects with obesity. Values are number (%) or means ± SD.

	WL *n* = 28	NWL *n* = 27	*p* Value
Age	11.2 ± 2.7	11.4 ± 2.8	ns
Female	13(44.8%)	13(48.1%)	ns
Prepubertal	11(40.7%)	9(31.0%)	ns
Height (cm)	149.5 ± 13.7	151.6 ± 17.0	ns
Height (SDS)	1.0 ± 1.1	1.1 ± 1.2	ns
Weight (kg)	67.3 ± 21.0	75.2 ± 25.4	ns
BMI (kg/m^2^)	29.3 ± 5.2	31.8 ± 5.7	ns
BMI z-score	2.23 ± 0.51	2.50 ± 0.57	ns
Waist (cm)	88.9 ± 14.2	95.1 ± 13.7	ns
SBP (mmHg)	123.3 ± 18.2	126.0 ± 15.9	ns
DBP (mmHg)	78.0 ± 11.8	77.7 ± 10.0	ns

Legend: *n*, number of subjects; ns, not significant; SDS, standard deviation score; BMI, body mass index; SBP, systolic blood pressure; DBP, diastolic blood pressure.

**Table 2 nutrients-13-01301-t002:** Changes in echocardiographic and vascular variables during the 12-months behavioral intervention in weight losers (WL) and non-weight losers (NWL) children with obesity. Values are number (%) or means ± SD.

	WL *n* = 28			NWL *n* = 27		
	T0	T12	*p* Value	T0	T12	*p* Value
HR (b/min)	81.1 ± 12.8	73.8 ± 10.3	0.01	85.8 ± 10.7	84.7 ± 12.7	ns
EF (%)	69.8 ± 7.9	71.1 ± 7.9	ns	70.0 ± 9.9	69.4 ± 6.8	ns
FS (%)	39.7 ± 6.3	40.5 ± 4.6	ns	42.9 ± 7.4	40.1 ± 5.7	ns
LVEDD (mm)	45.4 ± 6.0	47.1 ± 6.6	ns	46.1 ± 7.7	47.0 ± 5.1	0.04
LVEDD z-score	−0.65 ± 0.91	−0.24 ± 0.91	ns	−1.35 ± 1.28	−0.94 ± 1.19	ns
LVESD (mm)	27.5 ± 4.8	28.0 ± 5.1	ns	28.2 ± 7.1	28.4 ± 3.4	ns
LVESD z-score	−0.74 ± 1.07	−0.52 ± 1.13	ns	−1.16 ± 1.13	−1.03 ± 1.24	ns
IVSD (mm)	7.6 ± 1.6	7.6 ± 1.7	ns	7.7 ± 2.0	8.2 ± 1.9	ns
IVSD z-score	−0.28 ± 0.94	−0.49 ± 0.80	ns	−0.09 ± 1.05	0.30 ± 1.06	ns
LVPWD (mm)	7.5 ± 1.8	7.5 ± 1.2	ns	8.5 ± 2.8	7.7 ± 2.1	ns
LVPWD z-score	0.20 ± 0.20	0.31 ± 1.00	ns	0.46 ± 0.95	−0.25 ± 1.08	0.004
LAD (mm)	32.0 ± 5.2	33.0 ± 4.9	ns	33.7 ± 6.1	33.4 ± 5.4	ns
LAD z-score	1.12 ± 1.25	0.83 ± 1.22	ns	1.26 ± 1.00	1.54 ± 0.85	ns
Ao (mm)	24.0 ± 3.2	25.0 ± 3.8	ns	2.5 ± 0.5	2.65 ± 0.41	ns
LA/Ao ratio	1.40 ± 0.21	1.34 ± 0.22	ns	1.35 ± 0.22	1.30 ± 0.19	ns
LV mass (g)	113.1 ± 47.2	123.2 ± 40.9	ns	131.7 ± 85.1	121.0 ± 53.2	ns
LVmass index (g/m^2^)	66.8 ± 18.3	73.9 ± 17.8	ns	72.0 ± 30.9	64.8 ± 17.5	ns
LV mass z-score	−0.001 ± 1.479	0.11 ± 1.17	ns	0.26 ± 1.49	−0.15 ± 1.07	ns
RWT	0.33 ± 0.07	0.33 ± 0.07	ns	0.37 ± 0.09	0.32 ± 0.08	0.017
LV area (cm^2^)	26.5 ± 6.0	26.9 ± 5.7	ns	26.9 ± 7.2	26.5 ± 5.0	ns
LV volume (mL)	83.0 ± 27.3	84.8 ± 26.9	ns	86.3 ± 36.9	83.5 ± 23.4	ns
LA area (cm^2^)	12.9 ± 3.5	14.4 ± 2.8	0.04	13.9 ± 4.6	13.9 ± 3.1	ns
LA volume (mL)	32.5 ± 13.9	37.2 ± 12.5	ns	31.6 ± 12.8	34.9 ± 12.2	ns
Mitral E (cm/s)	1.02 ± 0.20	1.12 ± 0.18	0.001	1.04 ± 0.24	1.21 ± 0.59	0.007
Mitral A (cm/s)	0.55 ± 0.14	0.56 ± 0.11	ns	0.60 ± 0.13	0.65 ± 0.23	ns
Mitral E/A ratio	1.91 ± 0.43	2.02 ± 0.44	ns	1.7 ± 0.44	1.8 ± 0.47	ns
CIMT (mm)	0.62 ± 0.17	0.52 ± 0.09	0.010	0.61 ± 0.11	0.55 ± 0.05	0.014
AoDd (mm)	10.0 ± 1.6	10.8 ± 1.7	0.031	10.8 ± 2.4	11.2 ± 1.6	ns
AoDs (mm)	12.5 ± 1.9	13.1 ± 1.5	ns	13.2 ± 2.3	13.5 ± 1.8	ns
Aortic Strain, S	0.25 ± 0.10	0.23 ± 0.12	ns	0.24 ± 0.15	0.21 ± 0.11	ns
Ep (mmHg)	201.8 ± 128.3	249.1 ± 161.5	ns	478.7 ± 1140.1	253.4 ± 149.2	ns
Ep*	2.7 ± 1.8	3.4 ± 2.2	ns	6.3 ± 15.5	3.0 ± 1.8	ns
BAD basal (mm)	3.40 ± 0.74	3.49 ± 0.54	ns	3.53 ± 0.73	3.44 ± 0.57	ns
BAD after (mm)	3.48 ± 0.68	3.38 ± 0.88	ns	3.70 ± 0.75	3.48 ± 0.94	ns
FMD%	3.4 ± 14.6	0.9 ± 10.5	ns	6.5 ± 18.4	1.8 ± 24.6	ns

Legend: *n*, number of subjects; HR, heart rate; ns, not significant; EF, LV ejection fraction; FS, LV fractional shortening; LVEDD, LV end-diastolic dimension; LVESD, LV end-systolic dimension; IVSD, interventricular septum diastolic dimension; LVPW, LV posterior wall diastolic dimension; LAD, LA end-systolic diameter; LA, left atrium; LV, left ventricle; RWT, relative wall thickness; E, peak velocity of early diastolic transmitral wave; A, peak velocity of late diastolic transmitral wave; CIMT, carotid intima-media thickness; AoDd, abdominal aortic diastolic diameter; AoDs, abdominal aortic systolic diameter; S, aortic strain; Ep, pressure strain elastic modulus; Ep*, pressure strain normalized for diastolic blood pressure; BAD brachial artery diameter; FMD, brachial artery flow-mediated dilation.

## Data Availability

The datasets generated during and/or analyzed during the current study are not publicly available but are available from the corresponding author on reasonable request.

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
