# Peer review of "Healthy Lifestyle Intervention and Weight Loss Improve Cardiovascular Dysfunction in Children with Obesity"

_nutrients, 2021, doi:10.3390/nu13041301_

Round 1
Reviewer 1 Report
The article describes a 12 months intervention study aiming at promoting a healthy lifestyle among children and adolescents with obesity. The intervention was based on diet and physical activity. For diet, the intervention consisted of the “administration” of a Mediterranean diet, an explanation of the Mediterranean food pyramid, an education on portion size and calories and a counseling education on food palatability. The physical activity intervention consisted of 45-60 min. daily sessions of aerobic training. The effect of the intervention was assessed through anthropometric variables, physiological and biochemical variables and through cardiac and vascular structures variables (measured by echocardiography and sonography). It is a before-after study but the comparison between groups does not follow this logic and the authors compare separately weight losers and non-weight losers. The objective to evaluate the impact of the intervention cannot be achieved with the comparisons performed in the presented study. The results presented are not in line with the objective of the study.
The major weakness of this study is that the objective announced is not in line with the methodology and the results presented. It is a very important scientific weakness and I would recommend not to publish the article as submitted. I recognize that there might be interesting data for the readers included in this study but they should be presented in a totally other manner to be published with scientific soundness.
The sample size calculation was made to detect a 10% change in the left ventricular diameter between weight-losers and non-weight-losers. However, the researchers enrolled 55 subjects. This, in my opinion, illustrates the fact that the objective of the study is unclear.
Other important weaknesses are:
- the poor and unclear definition of the intervention. I would advise to have a look on recommendations for intervention reporting (ex. TIDIER reporting guidelines: https://www.equator-network.org/reporting-guidelines/tidier/).
- Results and statistics: presence of multiple testing that is a bad practice.
For a deep revision of this manuscript, I would suggest to present the data on cardiovascular structure only and to describe very clearly that it is a secondary analysis from a before-after study among the weight-losers subgroup. I insist though on the importance to describe very clearly and transparently what is analyzed, not to present a selection of positive results and not go beyond the results in the conclusion. The presentation of such results (completly revised manuscript) should not be presented as high evidence but as preliminary data to go further with other study designs.
General commentary on references : as references are not numbered in the references list but are indicated as number in the text, it is impossible to verify if the references are cited in the right place and if they are adequate. I guess the first reference in the list is number 1 in the text but not sure.
Author Response
We are glad to be invited to resubmit our manuscript and are thankful to the reviewers for their constructive comments.
Reviewer 1
1) It is a before-after study but the comparison between groups does not follow this logic and the authors compare separately weight losers and non-weight losers. The objective to evaluate the impact of the intervention cannot be achieved with the comparisons performed in the presented study. The results presented are not in line with the objective of the study.
The major weakness of this study is that the objective announced is not in line with the methodology and the results presented.
The sample size calculation was made to detect a 10% change in the left ventricular diameter between weight-losers and non-weight-losers. However, the researchers enrolled 55 subjects. This, in my opinion, illustrates the fact that the objective of the study is unclear.
Response: Thank you for your instructing comment. We agree with you and according to your comment we changed the title and the aim of the study. It is well known that behavioural therapy based on diet and physical activity is effective in reducing weight in children and adolescents with obesity (Henry BW, Ziegler J, Parrott JS, Handu D. Pediatric Weight Management Evidence-Based Practice Guidelines: Components and Contexts of Interventions. J Acad Nutr Diet 2018; 118(7):1301-11.e23.) thus we considered weight loss as a proxy of the compliance to the intervention.
Changes to the text:
Title: “Healthy lifestyle intervention and weight loss improve cardio-vascular dysfunction in children with obesity”.
Introduction: “Therefore, the purpose of this study was to determine the effects of weight loss, after a 12-month healthy-lifestyle intervention based on diet plus physical activity, on cardiovascular structure and function in children and adolescents with obesity”.
2) The poor and unclear definition of the intervention. I would advise to have a look on recommendations for intervention reporting (ex. TIDIER reporting guidelines: https://www.equator-network.org/reporting-guidelines/tidier/).
Response: Thank you for suggestion. We modified the section of the intervention description.
Changes to the text:
Materials and Methods, Intervention: “Moreover, subjects underwent a physical activity regimen. It was suggested to perform daily session of aerobic training for 45-60 minutes per day for 7 days per week. The type of training was explained by the pediatric endocrinologist and included aerobic activities fast walking, running, ball games, or swimming. Subjects could choose the type of activity based on their preference and their age”.
3) Results and statistics: presence of multiple testing that is a bad practice.
Response: Thank you for your comment. We used the test that was more appropriate for the analysis, the Wilcoxon signed-rank test was used to assess changes in cardiovascular variables and a two-way repeated measure ANOVA was performed to evaluate the time effect, the treatment effect and the interaction effects.
4) For a deep revision of this manuscript, I would suggest to present the data on cardiovascular structure only and to describe very clearly that it is a secondary analysis from a before-after study among the weight-losers subgroup. I insist though on the importance to describe very clearly and transparently what is analyzed, not to present a selection of positive results and not go beyond the results in the conclusion. The presentation of such results (completly revised manuscript) should not be presented as high evidence but as preliminary data to go further with other study designs.
Response: Thank you for your suggestion. According to your comment we omitted anthropometric and biochemical data and we deleted table 2. In the discussion and conclusion we specified several times that these are preliminary data and further larger studies and RCT are warrant to confirm our findings.
5) General commentary on references: as references are not numbered in the references list but are indicated as number in the text, it is impossible to verify if the references are cited in the right place and if they are adequate. I guess the first reference in the list is number 1 in the text but not sure.
Response: We are sorry that you did not see numbers next to references. In our version we saw them. We hope that in the revised version of the manuscript there will be no such problem.
Reviewer 2 Report
Aim of this study was to determine the effects of a 12-month healthy life-style intervention based on diet plus physical activity on cardiovascular structure and function in children and adolescents with obesity.
The work is well designed and well conducted by the authors.
This is a longitudinal study, which provides greater robustness to the research.
The topic is not new but it reinforces the importance of a healthy lifestyle, combining diet and physical activity.
The great contribution is the number of parameters analyzed, which allows a very complete vision of the intervention.
The establishment of two groups based on weight loss and the comparison between the two is interesting, but the global presentation of the results, at T0 and T12, could also be considered. Especially since the title does not mention the comparison between groups, and that the groups would be determined based on the results of the intervention.
Lines 82-84: The physical activity carried out is explained, that is, 45-60 minutes daily sessions of aerobic training (fast walking, 83 running, ball games, or swimming). It is suggested that it be explained and specified with greater precision if some type of control of intensity, load, pauses, etc. was carried out, since this variable seems poorly controlled. Was physical activity really daily for 7 days a week and for 12 months? If so, it must be specified.
The analysis of the other variables and the statistical analysis are correct, and the results support the conclusion. In my opinion, the authors can improve the section of conclusions, specifying certain findings better, and not presenting it as a single conclusion referring to lifestyle.
The limitation section is appreciated and considered, although the authors should include, among others, the control of physical activity.
Finally, and in such a current topic, it would be advisable to review the bibliographic references, many of them more than 10 years old.
Author Response
We are glad to be invited to resubmit our manuscript and are thankful to the reviewers for their constructive comments.
Reviewer 2
1) The establishment of two groups based on weight loss and the comparison between the two is interesting, but the global presentation of the results, at T0 and T12, could also be considered. Especially since the title does not mention the comparison between groups, and that the groups would be determined based on the results of the intervention.
Response: Thank you for your comment. We have decided to perform this analysis since on the whole population there were not significant differences in the majority of parameters between T0 and T12. Our explanation has been that many patients had not lose weight after the behavioural intervention and we decided to reanalyse data on the basis of the weight loss as a proxy of the compliance to the program. We changed the title in: “Healthy lifestyle intervention and weight loss improve cardiovascular dysfunction in children with obesity”.
2) Lines 82-84: The physical activity carried out is explained, that is, 45-60 minutes daily sessions of aerobic training (fast walking, running, ball games, or swimming). It is suggested that it be explained and specified with greater precision if some type of control of intensity, load, pauses, etc. was carried out, since this variable seems poorly controlled. Was physical activity really daily for 7 days a week and for 12 months? If so, it must be specified.
The limitation section is appreciated and considered, although the authors should include, among others, the control of physical activity.
Response: Thank you for your comment. During the counselling the pediatric endocrinologist and the nutritionist explained which type of training was better, based on the subject’s preference and his age, for example fast walking for younger children and ball games for older one and it was suggested to practice PA daily, for 7 days a week. We added this information: Materials and methods, intervention: “Moreover, subjects underwent a physical activity regimen including 45-60 minutes daily sessions of aerobic training (fast walking, running, ball games, or swimming). The regimen provided aerobic training for 7 days per week”.
We have not monitored the exercise intensity and the effective duration and this has been added to the limits of this study, as you suggested: Discussion, limitations of the study: “Furthermore, regarding the intervention, we have not monitored the intensity and the effective duration of physical activity. The use of telemetric methods like smart watches could be helpful to monitor the subjects’ compliance”.
3) In my opinion, the authors can improve the section of conclusions, specifying certain findings better, and not presenting it as a single conclusion referring to lifestyle.
Response: We have improved the conclusions, according to your suggestion.
4) Finally, and in such a current topic, it would be advisable to review the bibliographic references, many of them more than 10 years old.
Response: Thank you for your comment. We have updated the bibliography.